environmental science/atmospheric science/biophysics

urban area, land use land cover, land surface temperature, urban sprawl, surface urban heat island

**Author for correspondence:**
Emmanuel Olaoluwa Eresanya
e-mail: eresanyaemmanuel44@gmail.com,
emmanuel44@scsio.ac.cn

# Investigation of the changing patterns of the land use land cover over Osogbo and its environs

Emmanuel Olaoluwa Eresanya[1,2],
Mojooluwa Toluwalase Daramola[3],
Olufemi Sunday Durowoju[3,4] and Peace Awoyele[3]

[1]Department of Marine Science and Technology, Federal University of Technology, P.M.B 704, Akure, Nigeria
[2]South China Sea Institute of Oceanology, Chinese Academy of Sciences, 164 West Xingang Road, Guangzhou 510301, People's Republic of China
[3]Department of Meteorology and Climate Science, Federal University of Technology, P.M.B 704, Akure, Nigeria
[4]Department of Geography & Resource Management, Osun State University, Okokum, Osun State, Nigeria

 EOE, 0000-0002-5942-0849

The progressive nature of urbanization plays a prominent role in land–atmosphere processes, which have corresponding impacts on the general environment. This research investigated the changing patterns of the land use land cover over Osogbo and its environs using remote sensing data obtained from Landsat TM, ETM+ and OLI/TIRS sensors. The changes in four land use classes were assessed for the years 1984, 2000 and 2015. The land surface temperature (LST) of the area was estimated from the satellite images covering the study periods, and the surface urban heat island (SUHI) process was also investigated between the city of Osogbo and the surrounding towns. The results showed major urban expansion leading to urban sprawl within the vicinity. Urban area increased by 5106 ha while vegetation decreased by 8653 ha between 1984 and 2015 indicating major variations in the land surface features. This was revealed by the increase in the LST over the locations which ranged between 22.6°C and 30°C (mean, 25.2°C) in 1984 and between 29.3°C and 36.7°C (mean, 31°C) in 2015. The highest SUHI intensity was observed between the major urban area (Osogbo) and least developed towns. The continuous increase in the surface temperature of the environment due to the continuous variations in the land surface properties implies increased risk of heat-related environmental issues such as deterioration of thermal comfort conditions.

# 1. Introduction

The world is undergoing the most tremendous and influential urban growth in the history of mankind. Earlier before now, the world population was concentrated primarily in rural areas but by 2014, about 54% of the world's population reside in urban areas such as towns and cities, which is expected to increase to about 66% in the coming decades [1]. United Nations (UN) Centre for Human Settlements [2] predicted that by 2030, about 5 billion people would be residing in urban areas; this is expected to influence and have serious impacts on the social–economic and environmental transformations across the continent of Africa and Asia. Sterling & Ducharne [3] estimated that about 40% of the Earth's surface is covered by human-induced (anthropogenic) land cover and that impervious surfaces are systematically replacing the natural vegetation-dominated landscapes.

The prompt continuous expansion and growth occurring in major cities of developing countries play effective roles in the environment, causing changes in ecological processes on varying scales (local, regional and global). Urbanization has not only been beneficial to people but has also triggered severe impacts on the environment and economy. Its benefits include easy and quick access to basic social amenities such as quality education, medical services, electricity and water supply, and better economic and job opportunities, while its severe impacts may include forfeiture of agricultural land [4], habitat destruction which breaks down ecological cycles [5,6], the effects of urban heat island (UHI) [7–9], and contamination of air, soil and water [10]. Due to built-up environment, cities create their own microclimate, which generally enhances the effects of global climate change.

Awareness of the negative impacts of habitat destruction or conversion has been on the increase; as such, it is becoming known that the populations, infrastructure and ecology of cities are at risk from the impacts of climate change-related hazards. The waves of different land use, for example, in urban areas, can also lead to an increase in temperature at the city centre compared to the surrounding countryside (villages and less-developed suburbs), known as the UHI effect. There are numbers of considerable factors prompting the formation of UHI, e.g. characteristic of urban materials, urban geometry, anthropogenic heat, weather condition and geographic location. Urban roofs covering material, urban structure, street width, directions, etc., collectively contribute towards UHI. UHI is characterized by a great expanse of non-evaporating and impermeable materials covering most of urban areas with a far-reaching upsurge in sensible heat flux at the detriment of latent heat flux [11,12]. There are distinct differences in the behavioural pattern of natural surfaces and urban surfaces relating to the absorption of short-wave and long-wave radiations, evaporation, release of man-made heat and the blocking of the prevailing wind [13].

Several studies considered the advantages of remote sensing technology in investigating the changes in land surface properties over some areas in Nigeria [14–19]. These studies were over different locations in Nigeria such as Akure [16], Ibadan [18] and Sokoto [19]. However, there is no study at present that provides information on the changing pattern and intensity of the land surface in Osogbo, despite the increase in the population and rapid industrialization that have taken over the town, except Aguda & Adegboyega [20]. However, the study of Aguda & Adegboyega [20] focused only on the land use changes over Osogbo and not the surrounding towns. This study seeks to investigate the changes in the land use land cover, as well as the corresponding impact on the thermal condition over the land surface of Osogbo and its environs. In view of this, the land surface temperature (LST) of the area was estimated from the satellite images covering the study periods. Furthermore, the surface urban heat island (SUHI) phenomenon was assessed across the city of Osogbo and the surrounding towns.

# 2. Data and methods

## 2.1. Study area

This research is carried over the region of Osogbo (7°48″ N, 4°35″ E), Osun state, Nigeria (figure 1). Osogbo, the capital of Osun state, is located in the southwestern region of Nigeria. Osogbo city has boundaries with Ede, Olorunda, Obokun, Egbedore, Orolu, Ifelodun, Boripe, Atakumosa and Ilesa. According to Köppen classification, it falls under the tropical wet and dry climate (Aw). The climate is influenced by the maritime tropical (mT) air mass, the continental tropical (cT) air mass and the equatorial easterlies. The average annual temperature is 26.1°C, and the annual precipitation is 1241 mm, which falls mostly in summer. The driest and wettest months are January and September with 9 and 202 mm values, respectively, while the warmest and coldest months are March and

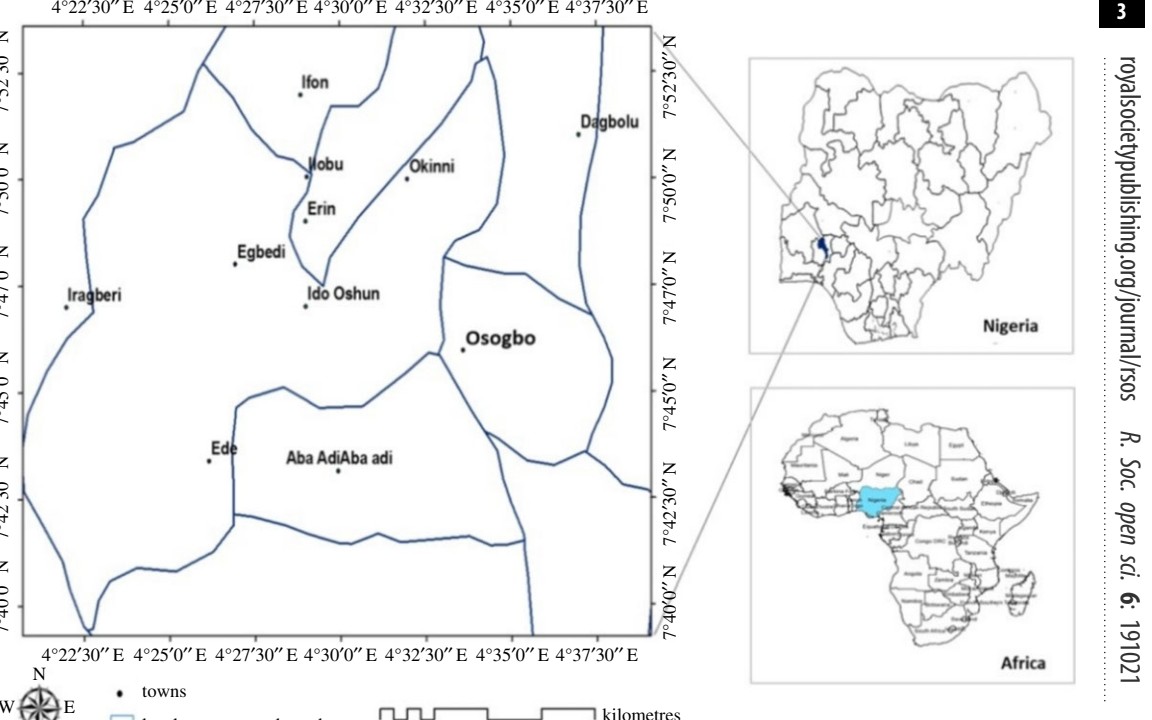

**Figure 1.** Map of the study region showing the towns considered in the study.

August with an average 28.3°C and 23.7°C, respectively. Precipitation varies with 193 cm between the driest and the wettest months. The variation in annual temperature is about 4.6°C. Osogbo is the administrative and commercial centre of Osun state. Osogbo has been witnessing tremendous increase in population since the creation of Osun state but this city has witnessed increase in road expansion and construction since the advent of the present government (2011 till date), which had led to increase urbanization and population inflow into Osogbo and its suburbs.

## 2.2. Data

Cloud-free satellite data from the Landsat sensors TM, ETM+ and OLI/TIRS were used in this study. The images were obtained from the United States Geological Survey (USGS) for 1984, 2000 and 2015, covering a period of 31 years. The details of the data are given in table 1.

## 2.3. Image classification

The acquired images were analysed and processed, and then geometrical correction was carried out on the images. The images were referenced to the Universal Transverse Mercator (UTM) Zone 31 North and projected to World Geodetic System (WGS) 84. Subsequent to geometric and radiometric corrections of satellite images, caution was taken to prepare and extract land use maps using the supervised classification method into four classes: barren land, vegetation, water body and urban, as shown in table 2. The classification scheme was developed based on the knowledge of the research region. Several band combinations were produced for the images so as to adequately depict the land use land cover classes, and the supervised classification was also carried out on the images by applying the classification scheme. Maximum likelihood parametric rule, which is the most extensively used parametric classification algorithm [21,22], was adopted for the image classification.

The accuracy assessment was done for the classified images. This was achieved using the *κ*-coefficient through the generation of 140 random points across the images. The points were compared with the referenced dataset from the study area.

**Table 1.** Details of the satellite images used for the research.

| satellite | sensor | path | row | source | date acquired | resolution (m) | time (West African) |
|---|---|---|---|---|---|---|---|
| Landsat5 | TM | 190 | 55 | USGS | 11 Dec 1984 | 30 | 10.26 |
| | | | | | | 120 (30) | |
| Landsat7 | ETM+ | | | | 15 Feb 2000 | 30 | 10.49 |
| | | | | | | 60 (30) | |
| Landsat8 | OLI/TIRS | | | | 17 Dec 2015 | 30 | 10.56 |
| | | | | | | 100 (30) | |

**Table 2.** Classification scheme used in the study.

| land use land cover class | description |
|---|---|
| barren land | bare surface with no vegetation, open land sparse shrubs mixed with bare soil |
| vegetation | forested areas, densely vegetated areas, trees, grassland, crops, gardens and parks |
| water body | lakes, rivers, reservoirs, ponds |
| urban | built-up, settlement, residential, commercial and industrial areas |

## 2.4. Computation of land surface temperature

The first step in computing the LST was the conversion of the digital number (DN) to spectral radiance

$$L_\lambda = \left(\frac{L_{\mathrm{MAX}} - L_{\mathrm{MIN}}}{Q_{\mathrm{CALMAX}} - Q_{\mathrm{CALMIN}}}\right) \times (\mathrm{DN} - Q_{\mathrm{CALMIN}}) + L_{\mathrm{MIN}} \tag{2.1a}$$

and

$$L_\lambda = (M_{\mathrm{L}} \times Q_{\mathrm{Cal}}) + A_{\mathrm{L}}, \tag{2.1b}$$

where $L_\lambda$ is the spectral radiance, DN is the digital number of each pixel, $L_{\mathrm{MAX}}$ and $L_{\mathrm{MIN}}$ are calibration constants, $Q_{\mathrm{CALMAX}}$ and $Q_{\mathrm{CALMIN}}$ are the highest and lowest range of values for rescaled radiance in DN obtained from the metadata, $M_{\mathrm{L}}$ is the band-specific multiplicative scaling factor, $A_{\mathrm{L}}$ is the band-specific additive rescaling factor. Equations (2.1a) and (2.1b) apply to Landsat 4/5 and Landsat 8, respectively.

This was followed by the conversion of the spectral radiance to at-sensor brightness temperature ($T_{\mathrm{B}}$) using Planck's inverse function.

$$T_{\mathrm{B}} = K_2 \ln\left(\frac{K_1}{L_\lambda} + 1\right)^{-1}, \tag{2.2}$$

where $K_1$ and $K_2$ are calibration constants for the landsat image given as $K_1 = 607.76$, 666.09 and 774.89, while $K_2 = 1260.56$, 1282.71 and 1321.08 (W m$^{-2}$ sr$^{-1}$ μm$^{-1}$) for Landsat TM, ETM+ and OLI/TIRS, respectively, and $L_\lambda$ is the spectral radiance for the thermal band in (W m$^{-2}$ sr$^{-1}$ μm$^{-1}$).

The LST was estimated using the equation below

$$\mathrm{LST}_k = T_{\mathrm{B}}\left(1 + \left[\frac{L_\lambda \times T_{\mathrm{B}}}{\rho}\right] \times \ln \varepsilon\right)^{-1}, \tag{2.3}$$

where $\mathrm{LST}_k$ is land surface temperature (in Kelvin), $L_\lambda$ is the wavelength of emitted radiance which equals 11.5 μm, $\rho$ is $h \times c/\sigma$ ($1.438 \times 10^{-2}$ m K), $h$ is Planck's constant ($6.26 \times 10^{-34}$ J s), $c$ is the velocity of light ($2.998 \times 10^8$ m s$^{-1}$), $\sigma$ is Stefan Boltzmann's constant ($1.38 \times 10^{-23}$ J K$^{-1}$) and $\varepsilon$ is the emissivity.

The final step was the conversion of the LST derived from equation (2.3) to LST in degree Celsius

$$\mathrm{LST}_{\mathrm{C}} = \mathrm{LST}_{\mathrm{K}} - 273.14. \tag{2.4}$$

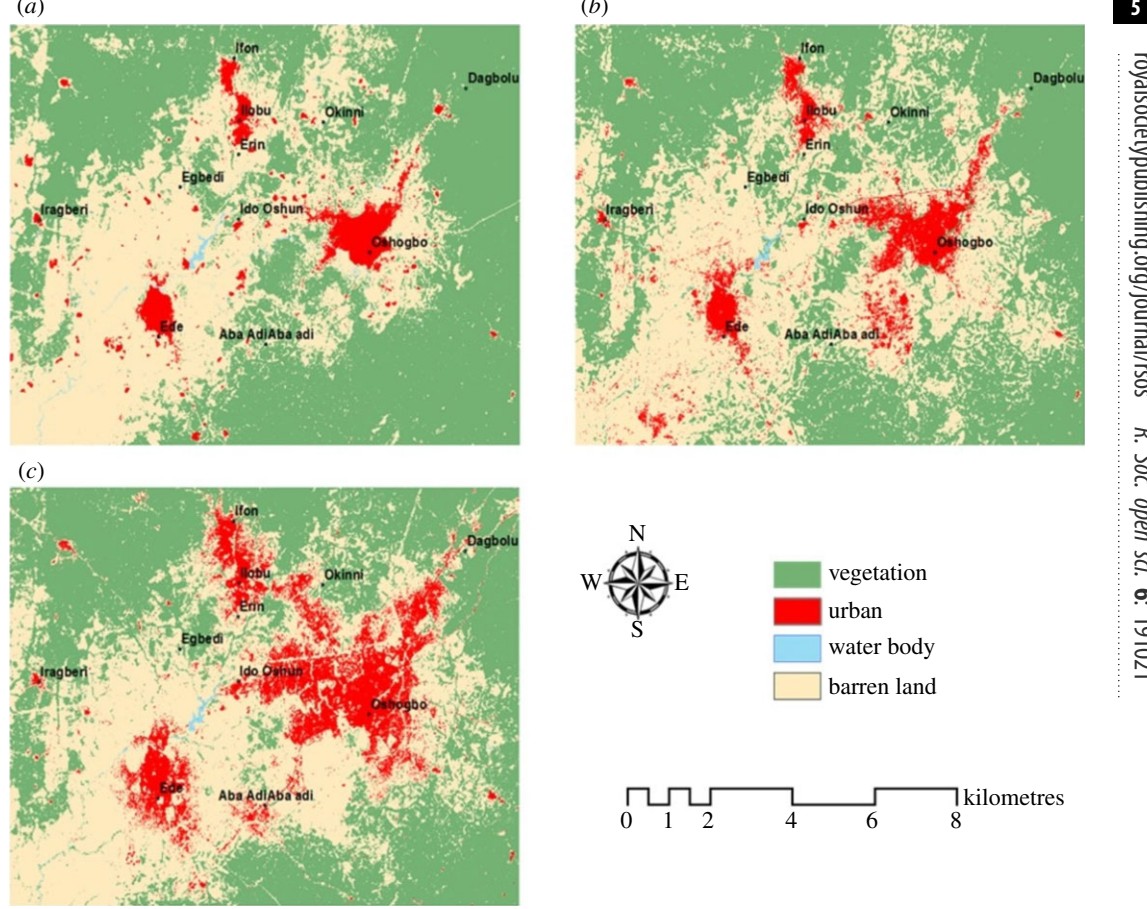

**Figure 2.** Land use land cover map of the region of study for (a) 1984, (b) 2000 and (c) 2015.

## 2.5. Surface urban heat island estimation

The SUHI which represents the LST difference between an urban area and a suburban/rural area was computed using equation (2.5). Osogbo city was chosen as the urban centre while the other towns surrounding the city were classified as suburban/rural.

$$\text{SUHI} = \text{LST}_{\text{urban}} - \text{LST}_{\text{rural}}. \tag{2.5}$$

Several other methods of SUHI computations are available in the literature [23,24]; however, this study employed the use of the mean values of the LST of the urban area and surrounding rural areas.

# 3. Results and discussion

## 3.1. Land use land cover analysis

The land use maps of 1984, 2000 and 2015 are shown in figure 2. In 1984, the only noticeable urban areas were Osogbo, Ede and the Ilobu axis, with Osogbo been the largest urban area. In 2000, these urban areas witnessed increase as the area expanded; however, the most noticeable observation is the increase in urban areas in 2015. At this period, the areas experienced dramatic increase as the expansion occurred, and other locations previously rural became urbanized. The expansion of Osogbo and Ilobu axis was such that urban sprawl, which is basically defined as the spreading of urban developments, occurred between 1984 and 2015.

The accuracy of the classified images was assessed, and the result is presented in table 3. The overall accuracy of the classification was 84.2% for 1984, 84.35% for 2000 and 84.14% for 2015. The $\kappa$-coefficient was 0.803, 0.812 and 0.806 for 1984, 2000 and 2015, respectively; the result reveals a reasonable classification accuracy.

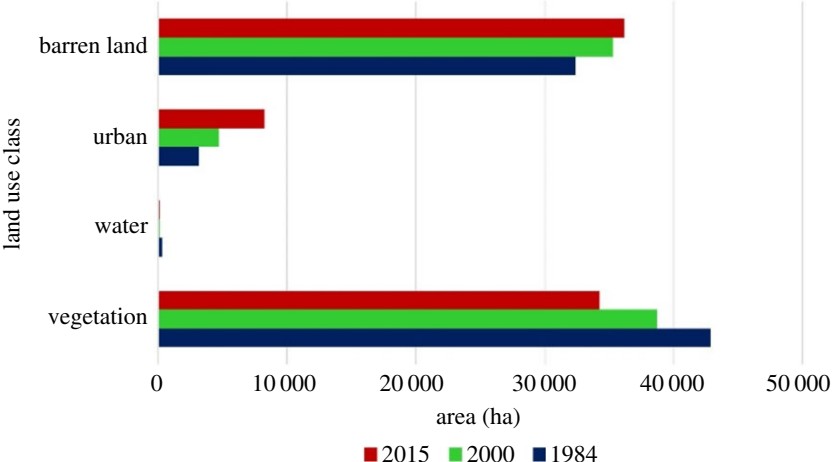

**Figure 3.** Area coverage for each land use class.

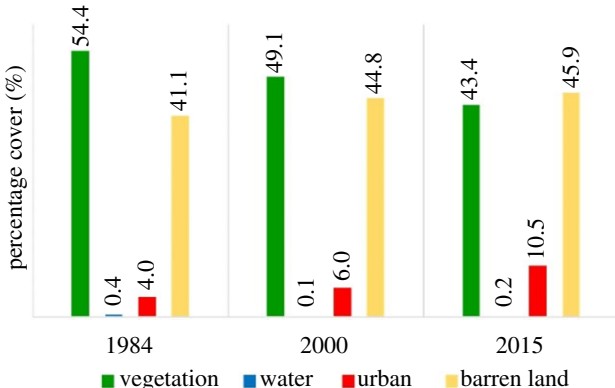

**Figure 4.** Percentage cover of each land use class for each year.

**Table 3.** Accuracy assessment.

|  | 1984 | 2000 | 2015 |
|---|---|---|---|
| overall classification accuracy | 84.2 | 84.35 | 84.14 |
| overall $\kappa$-statistics | 0.803 | 0.812 | 0.806 |

In order to explain and show the changes in the land use land cover over Osogbo and its environs, the area coverage (hectares) of the land cover classes was analysed and plotted as shown in figure 3. In 1984, vegetation occupied the majority of the land cover with an area coverage of about 43 000 ha, barren land was about 32 500 ha while water and urban area were about 340 and 3200 ha respectively. In 2000, vegetation reduced to about 39 000 ha, while the barren land and urban area increased to about 35 300 and 4800 ha, respectively. In 2015, the area coverage of vegetation was about 34 300 ha, barren land was about 36 200 ha and urban area was about 8300 ha.

Figure 4 reveals the percentage cover of each land use class for each of the years studied. As stated earlier, the area coverage of vegetation decreased over time, from 54.4% in 1984 to 49.1% in 2000 and finally 43.3% in 2015. Urban and barren land increased from 4.0% and 41.1% in 1984 to 6.0% and 44.8% in 2000, and then to 10.5% and 45.9% in 2015, respectively. Water body was 0.4%, 0.1% and 0.2% in 1984, 2000 and 2015, respectively. It is evident that vegetation cover occupied a majority of the land surface area within the region; however, the percentage decreased towards 2015, with barren land and urban increasing. Figure 3 also showed that the expansion of the urban area cover which was accompanied by increase in barren land brought about conversion of the vegetated portion of the land use.

Figure 5 shows the change in area cover for each land use between the different intervals. Between 1984 and 2000, vegetation lost about 4200 ha to urban and barren land while urban and barren land gained 1558 and 2861 ha, respectively. From 2000 to 2015, vegetation lost another 4456 ha to other

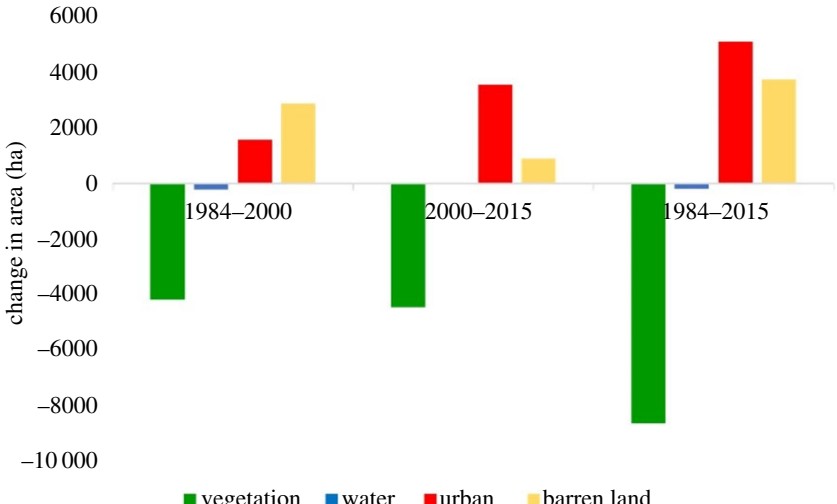

**Figure 5.** Change in area coverage for each land use class at different intervals.

land use classes, while both urban and barren land gained 3547 and 879 ha, respectively. The total loss of vegetation cover area over the region between 1984 and 2015 was 8653 ha, while the total increase in urban and barren land was 5105 and 3739 ha, respectively.

## 3.2. Land surface temperature analysis

This section presents the result of estimated LST from satellite images over the region. The LST maps are presented in figure 6. The LST distribution clearly shows the highest LST intensities over the urbanized areas of the region such as Osogbo, Ede and Ilobu axis. The lowest LST intensities are observed over the peripheries of the region designated as vegetation.

Figure 7 shows the LST over each of the locations considered in this study. Evident increase in the LST over the 31-year period is presented in the chart. The LST intensity over all the locations increased significantly between 1984 and 2015 with a mean value of 5.8°C. The highest increase within the period was over Osogbo with an increase of 6.8°C. The mean LST increased from 26.2°C in 1984 to 30.5°C in 2015 (table 4), indicating a progressive warming of the region.

## 3.3. Surface urban heat island

The LST has been recognized as a pointer to the existence of SUHI. The SUHI estimated from the LST over the region is presented in figure 8. SUHI was noted across the towns for each of the years studied. The mean SUHI intensity showed that Dagbolu, Okinni and Aba Adi had the highest intensities while Ilobu had the lowest intensity. The towns with the highest SUHI intensities were the rural areas of the region with lower intensity of urbanization and associated human activities, which bring about increased thermal intensity. Alternatively, Ilobu with the lowest intensity of SUHI is a suburban area.

# 4. Discussion

## 4.1. Land use land cover change

The result from this study reveals remarkable changes in the land use land cover of Osogbo and its environs over the years studied. Agricultural activities are known to be the major activities of the residents of Osogbo and its environs in the past, and about 68% of the land was used for agriculture-related practices; but since it became the state capital in 1991, the area had experienced a drastic change in land use. Most agricultural lands had been altered to built-up areas. Aside from the state capital institution of Osogbo in 1991, another noticeable factor responsible for urban spatial expansion was the establishment of Osogbo machine tools and steel rolling mills companies, which played a significant and pivotal role in the economic activities of the capital city and caused development along the

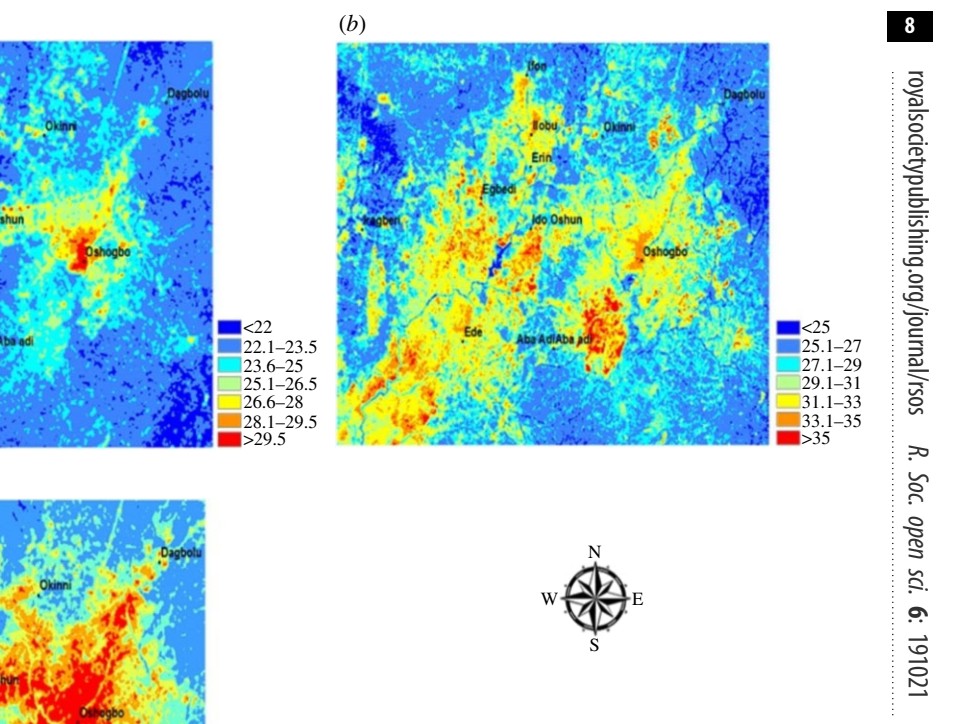

**Figure 6.** LST (°C) for (*a*) 1984, (*b*) 2000 and (*c*) 2015.

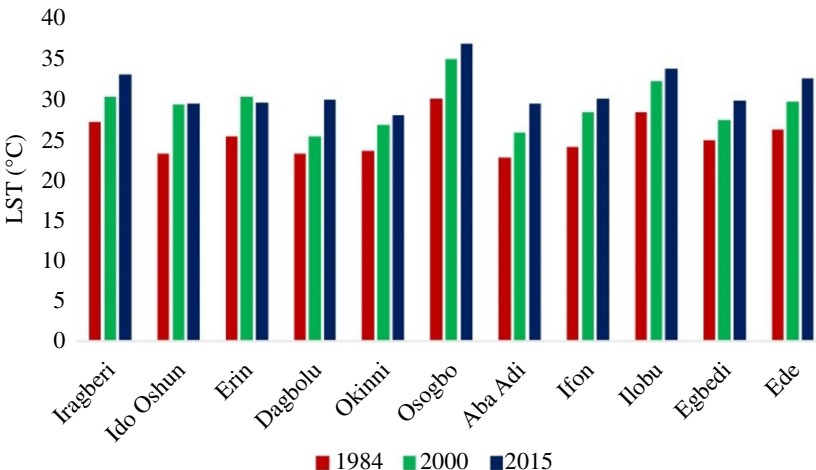

**Figure 7.** LST over Osogbo and the surrounding towns.

**Table 4.** Details of the LST for each year.

|  | minimum (°C) | maximum (°C) | mean (°C) |
| --- | --- | --- | --- |
| 1984 | 20.9 | 31.6 | 26.2 |
| 2000 | 22.7 | 36.6 | 29.7 |
| 2015 | 23.9 | 37.1 | 30.5 |

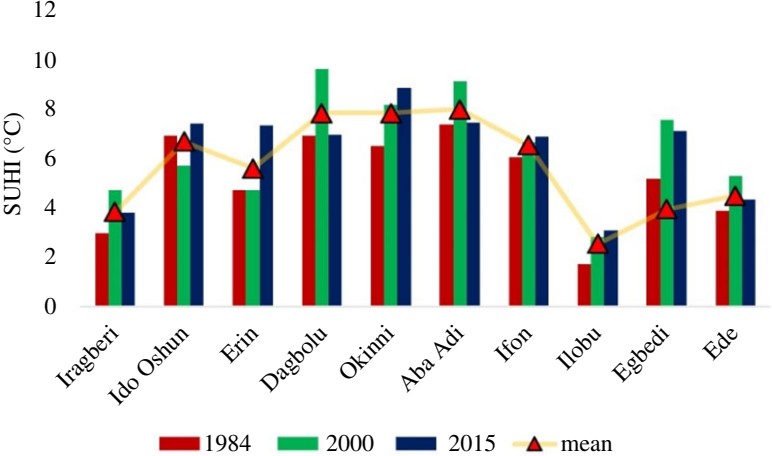

**Figure 8.** SUHI for each location.

northern axis of the city. The study region form was largely axial in its spatial expression, growing along major arteries of transportation. These axes of growth were separated by undisturbed forest, secondary vegetation and wetlands. This became most noticeable in 2000 which represents the beginning of noticeable development in the region, with a rise in the built-up regions and reduction in vegetative coverage, which is due to the massive influx of people from the suburbs to the capital. In 2000, the city of Osogbo experienced major developments, such as a new government secretariat built by the state government, and also other parastatals began to spring up, such as House of Assembly Complex and Central Bank Branch along Gbongan-Ibadan axial road. Private residential and commercial developments were also springing up in response to the burgeoning population of mostly government workers and their dependents, resulting in low-density development towards the periphery.

This study corroborates the findings of Samuel & Atobatele [25] which established the fact that transition to civil rule yielded economic dividends which further intensified urban expansion, with concomitant accelerated loss of both vegetal cover and water bodies in 2000. The period equally marked the implementation of a wage increase of 15% in the country which afforded workers the opportunity to build their own houses. The economic fortunes of the people have a significant effect on the trajectory of urban development coupled with the inflow of migrants from suburbs and the pastoral areas in search of government occupations and contracts in the newly created state. The sustenance of urban land use growth in this period is partly due to the economic boom that led to the oversaturation of the surrounding lower-order settlements such as Dagbolu, Kelebe, Ofatedo, Okinni, Abere, Ido Osun, Ilobu, Ifon and Iragberi.

Since then, the vacant land and wetlands within the urbanized areas were being developed, in response to increased demand from the government workers (both federal and state) who had to move to the state capital as a result of the creation of the state. Expectedly, the appreciable gain in urban expansion caused a concomitant reduction in both vegetation cover and water. As the residents in the city increase across the years, this is followed by a corresponding increase in the urbanized area with a consequential reduction in vegetative area. Land use land cover analysis of Osogbo in 2015 revealed that there is an increase in development with the further expansion in built-up across the area, which is expected to continue due to the continuous rise in the number of its residents. The inflow of people from outside the urban areas to the urban centres in search of job opportunities and better standard of living leading to increased urban expansion has also been noticed in some of the capital cities in Nigeria such as Lagos [15] and Ibadan [18], as well as in other countries [26–28].

By 2015, the area coverage of urban increased while vegetation reduced. During this period, the city appeared to have expanded to its traditional and administrative limits, as the development of hitherto undeveloped plots within the built-up areas, which had started earlier, intensified as identified by Aguda & Adegboyega [20]. It was at this period that the conversion of vegetation reached its peak, with the consequent deleterious effect on the ecosystems services. The continued increase in the expansion of Osogbo city caused a spillover into other smaller suburban centres that were not considered as part of the city. However, adjoining towns such as Owode Ede, Ilobu, Erin Osun, Ifon Osun, Oba-Ile, Oba-Oke, Dagbolu, Ido Osun and Ede continued to receive the spillover population and development from Osogbo.

## 4.2. Effect of land use land cover change on land surface temperature

The changing land use land cover of Osogbo and the surrounding towns have effects on the surface thermal condition of the area. This was observed in the LST distribution and intensity for the years studied. The varying intensity of the LST with land use land cover has been reported in studies across the globe [29–33]. The intensity of LST decreased outwards from the urbanized areas to the areas with less urban surfaces as the highest intensities are within the urban core. The dispersion of LST intensity from the urban core to the peripheries of the region clearly shows the temperature differential between the urban area and other land use land covers such as vegetation. It can also be seen that the intensity of the LST increased over the area with time. Initially in 1984, the high LST intensity was concentrated over Osogbo; however, with the expansion of urban developments outwards from the town centre, the intensity increased over the altered land use areas. As stated earlier, Osogbo is the major urban area within the region, and with the degree of urbanization, population and intensity of human-induced activities, over the past couple of years, the surface temperature of the city increased significantly. This indicates the effect of urbanization on the thermal condition of the environment. According to Barat *et al.* [34], the rapid development of metropolitan areas accounts for intensification of the temperature of the land surface.

Urbanization and its associated activities have brought about changes in the thermal situation of an area [35–38], greatly affecting the intensity of SUHI, as factors such as impervious surfaces, intense human activities and reduced vegetation cover all bring about increase in the urban temperature [39–41]. This consequently affects the thermal comfort condition of the region [42,43]. The subject of increased intensity of urban surface temperature has been reported in cities over the globe [44–47]. Several mitigation strategies for the increased urban temperature have been considered, of which increased greenery of the cities has been proposed by several studies [48–50]. Vegetation within the urban environment, such as forest reserves and urban parks, affects the surface temperature due to the reduction in surface heating by the incoming solar radiation [51–54]. Vegetation cover such as trees shields the land surface from direct heating and therefore brings about reduced temperature of the surface compared with that of impervious surfaces which characterize the urban area.

## 5. Conclusion

The population of urban areas is on the continual increase with continuous rural–urban migration. This implies the need for human settlement such as housing and other required services. These activities have resulted in the modification of the landscape of different areas. The change in the land use land cover over the region of Osogbo was assessed in this research. The changes in land cover properties majorly proliferate the urban built-up areas, and decrease in vegetation cover, which is a prominent issue in our world today, was noted in this research. The urban area was observed to increase progressively towards the outer parts of the region. The phenomenon known as urban sprawl was clearly seen within the region over time. The urban areas had expanded such that towns have been linked with each other. Urban settlements, which were a few hectares at earlier years, had increased significantly outwards towards other towns. The increase in urban settlements and the corresponding decrease in vegetation cover were shown to influence the thermal condition of the region. The LST increased over all the towns between 1984 and 2015 as a result of growing infrastructure and increased human activities. LST, which is an indicator for ascertaining SUHI, was applied in the study to further investigate the SUHI phenomenon. SUHI was observed to exist within the region, as Osogbo, the main urban area, had higher surface temperature intensity compared with that of the surrounding towns. The highest SUHI intensity was noted between Osogbo and the rural towns, which indicated the consequence of urbanization on the spatial distribution of temperature and SUHI intensity. The results of this research provide vital statistics about the state of the changing nature of the land surface within the region of Osogbo and its environs. It also presents valuable information about the thermal condition of the area with respect to the changing urban landscape. This will serve as significant information required for suitable urban planning. This study also draws light on the importance of vegetation cover in the context of mitigation of increased temperature across urban areas. With the continuous increase in the mean temperature regionally [54,55], vegetation cover such as forest reserves and urban parks can be introduced and incorporated into the urban planning configuration of the region.

Data accessibility. Cloud-free satellite data from the Landsat sensors TM, ETM+ and OLI/TIRS were used in this study. The images were obtained from the United States Geological Survey (USGS) for 1984, 2000 and 2015 covering a period of 31 years. Data can be accessed from https://earthexplorer.usgs.gov/.

Authors' contributions. E.O.E. conceived and designed the study and coordinated the analysis; M.T.D. and P.A. participated in the preliminary data collection and analysis; E.O.E. and M.T.D. carried out sequence alignments with further analysis and drafted the manuscript; O.S.D. carried out the first draft of literature review; E.O.E. proofread the complete manuscript before submission. All authors gave final approval for publication.

Competing interests. We declare we have no competing interests.

Funding. The authors received no funding for this study.

Acknowledgements. The authors appreciate the United States Geological Survey (USGS) for the provision of the dataset used in this research work. We also want to thank the anonymous reviewers who provided comments that substantially improved the manuscript.

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
