## [Reviewer comments · Royal Society Open Science]

Review History

RSOS-191021.R0 (Original submission)

Review form: Reviewer 1 (Julius Akinyoola)

Is the manuscript scientifically sound in its present form?

Yes

Are the interpretations and conclusions justified by the results?

Yes

Is the language acceptable?

Yes

Do you have any ethical concerns with this paper?

No

Have you any concerns about statistical analyses in this paper?

No

Recommendation?

Major revision is needed (please make suggestions in comments)

Comments to the Author(s)

Abstract page. Line 3

The author should correct the statement 'the land surface over Osogbo'. it is an incorrect statement.

line 5

the author should explain what he mean by this Thirty-one-year period because he only perform analysis on 1984, 2000 and 2015 year one after the other.

The author is expected to to use keywords nor abbreviation in the keyword section.

Page 5

I expected that the map projection in fig 1 should reveal climatological classification at local scale of the study area.

Page 8, 12 & 13

There are a lot of repetition here. the author should rewrite these pagesA to avoid repetition.

There is no acknowledgement refer to in this work.

There are also citations that were not inline with the write up

Review form: Reviewer 2

Is the manuscript scientifically sound in its present form?

Yes

Are the interpretations and conclusions justified by the results?

Yes

Is the language acceptable?

Yes

Do you have any ethical concerns with this paper?

No

Have you any concerns about statistical analyses in this paper?

No

Recommendation?

Major revision is needed (please make suggestions in comments)

Comments to the Author(s)

The study aim is to present the Research on "Investigation of the changing patterns of the land surface over Osogbo and its environs" The manuscript is presented clearly and nicely. However,

to have scientific merit, the following comments should be addressed. I would like to suggest Major revision for this paper.

1. The introduction is too much. Better to reduce it.
2. Figure 1 = pls add a scale bar and African continent for Figure 1
3. Pls add the time of the three images (Table 1)
4. What are the points behind the selection of the satellite images (Table 1)
5. How did you measure the accuracy assessment of the LULC? Add details related to accuracy assessments.
6. Add some more related past research related to the UHI intensity in the African region. The calculation is not clear. There is two main methods used to classify the UHI intensity. Refer to the latest paper for your understanding. Read these papers for more knowledge.

- Spatial Analysis of Surface Urban Heat Islands in Four Rapidly Growing African Cities
- Impact of Landscape Structure on the Variation of Land Surface Temperature in Sub-Saharan Region: A Case Study of Addis Ababa using Landsat Data (1986-2016)
- Impact of Urban Surface Characteristics and Socio-Economic Variables on the Spatial Variation of Land Surface Temperature in Lagos City, Nigeria
- Quantifying Surface Urban Heat Island Formation in the World Heritage Tropical Mountain City of Sri Lanka

7. Figure 2 and 6 - No scale bar or north arrow.
8. Figure 6 - use the same legend for three maps, and it will help to compare the changes
9. Figure 7 - witch locations.
10. Better to divide result and discussion into Result and discussion. All result need to discuss in the result sections.

Decision letter (RSOS-191021.R0)

23-Aug-2019

Dear Dr Eresanya,

The editors assigned to your paper ("Investigation of the changing patterns of the land surface over Osogbo and its environs") have now received comments from reviewers. We would like you to revise your paper in accordance with the referee and Associate Editor suggestions which can be found below (not including confidential reports to the Editor). Please note this decision does not guarantee eventual acceptance.

Please submit a copy of your revised paper before 15-Sep-2019. Please note that the revision deadline will expire at 00.00am on this date. If we do not hear from you within this time then it will be assumed that the paper has been withdrawn. In exceptional circumstances, extensions may be possible if agreed with the Editorial Office in advance. We do not allow multiple rounds of revision so we urge you to make every effort to fully address all of the comments at this stage. If deemed necessary by the Editors, your manuscript will be sent back to one or more of the original reviewers for assessment. If the original reviewers are not available, we may invite new reviewers.

Please note that the two reviews are in close agreement that the core topic of your paper is sound but that some considerable reorganisation is needed prior to publication.

- Data accessibility

If you wish to submit your supporting data or code to Dryad (<http://datadryad.org/>), or modify your current submission to dryad, please use the following link:
<http://datadryad.org/submit?journalID=RSOS&manu=RSOS-191021>

- Competing interests

- Authors' contributions

- Acknowledgements

- Funding statement

Kind regards,

Alice Power

Editorial Coordinator

on behalf of Dr Mark Smith (Associate Editor) and Jon Blundy (Subject Editor)

Associate Editor's comments (Dr Mark Smith):

Thank you for submitting "Investigation of the changing patterns of the land surface over Osogbo and its environs" [RSOS-191021] to Royal Society Open Science. I have received 2 reviews of your manuscript, which are included below and/or attached. As you can see, the reviews are in close agreement with both indicating that major revisions are needed before your paper can be accepted.

Both reviewers were excited about the results presented in this contribution; however, both also expressed reservations about the manuscript in its current form. The majority of these changes are presentational and require clearer communication of the scientific ideas and the reorganisation of the material to minimise repetition. I am therefore returning the paper to you so that you can make the necessary changes.

Reviewers' Comments to Author:

Reviewer: 1

Abstract page. Line 3

The author should correct the statement 'the land surface over Osogbo'. it is an incorrect statement.

line 5

the author should explain what he mean by this Thirty-one-year period because he only perform analysis on 1984, 2000 and 2015 year one after the other.

The author is expected to use keywords nor abbreviation in the keyword section.

Page 5

I expected that the map projection in fig 1 should reveal climatological classification at local scale of the study area.

Page 8, 12 & 13

There are a lot of repetition here. the author should rewrite these pagesA to avoid repetition.

There is no acknowledgement refer to in this work.

There are also citations that were not inline with the write up

Reviewer: 2

Comments to the Author(s)

The study aim is to present the Research on “Investigation of the changing patterns of the land surface over Osogbo and its environs” The manuscript is presented clearly and nicely. However, to have scientific merit, the following comments should be addressed. I would like to suggest Major revision for this paper.

1. The introduction is too much. Better to reduce it.
2. Figure 1 = pls add a scale bar and African continent for Figure 1
3. Pls add the time of the three images (Table 1)
4. What are the points behind the selection of the satellite images (Table 1)
5. How did you measure the accuracy assessment of the LULC? Add details related to accuracy assessments.
6. Add some more related past research related to the UHI intensity in the African region. The calculation is not clear. There is two main methods used to classify the UHI intensity. Refer to the latest paper for your understanding. Read these papers for more knowledge.

- Spatial Analysis of Surface Urban Heat Islands in Four Rapidly Growing African Cities
- Impact of Landscape Structure on the Variation of Land Surface Temperature in Sub-Saharan Region: A Case Study of Addis Ababa using Landsat Data (1986–2016)
- Impact of Urban Surface Characteristics and Socio-Economic Variables on the Spatial Variation of Land Surface Temperature in Lagos City, Nigeria
- Quantifying Surface Urban Heat Island Formation in the World Heritage Tropical Mountain City of Sri Lanka

7. Figure 2 and 6 – No scale bar or north arrow.
8. Figure 6 – use the same legend for three maps, and it will help to compare the changes
9. Figure 7 – witch locations.
10. Better to divide result and discussion into Result and discussion. All result need to discuss in the result sections.

Author's Response to Decision Letter for (RSOS-191021.R0)

See Appendix A.

Decision letter (RSOS-191021.R1)

06-Nov-2019

Dear Dr Eresanya:

On behalf of the Editors, I am pleased to inform you that your Manuscript RSOS-191021.R1 entitled "Investigation of the changing patterns of the land use land cover over Osogbo and its environs" has been accepted for publication in Royal Society Open Science subject to minor revision in accordance with the referee suggestions. Please find the referees' comments at the end of this email.

The reviewers and Subject Editor have recommended publication, but also suggest some minor revisions to your manuscript. Therefore, I invite you to respond to the comments and revise your manuscript.

- Ethics statement

- Data accessibility

<http://datadryad.org/submit?journalID=RSOS&manu=RSOS-191021.R1>

- Competing interests

- Authors' contributions

- Acknowledgements

- Funding statement

Because the schedule for publication is very tight, it is a condition of publication that you submit the revised version of your manuscript before 15-Nov-2019. Please note that the revision deadline will expire at 00.00am on this date. If you do not think you will be able to meet this date please let me know immediately.

Supplementary files will be published alongside the paper on the journal website and posted on

the online figshare repository (<https://figshare.com>). The heading and legend provided for each supplementary file during the submission process will be used to create the figshare page, so please ensure these are accurate and informative so that your files can be found in searches. Files on figshare will be made available approximately one week before the accompanying article so that the supplementary material can be attributed a unique DOI.

on behalf of Dr Mark Smith (Associate Editor) and Jon Blundy (Subject Editor)
openscience@royalsociety.org

Associate Editor Comments to Author (Dr Mark Smith):

Associate Editor

Comments to the Author:

Many thanks for this resubmission and the further detail in the rebuttal letter. I am confident that you have addressed the reviewers' concerns.

However, I note one or two minor presentational issues remain with the manuscript, in some cases as part of the corrections themselves.

I can see a section with "[insert equation here]" just before equation (4) and the first response to a reviewers' comment yields a sentence that makes no sense in the abstract.

If you could correct these minor errors and further check the manuscript for similar such errors, then I would be happy to accept the revised manuscript.

Author's Response to Decision Letter for (RSOS-191021.R1)

See Appendix B.

Decision letter (RSOS-191021.R2)

13-Nov-2019

Dear Dr Eresanya,

It is a pleasure to accept your manuscript entitled "Investigation of the changing patterns of the

land use land cover over Osogbo and its environs" in its current form for publication in Royal Society Open Science. The comments of the reviewer(s) who reviewed your manuscript are included at the foot of this letter.

on behalf of Dr Mark Smith (Associate Editor) and Jon Blundy (Subject Editor)
openscience@royalsociety.org

Appendix A

Dear Editor,

The authors wish to inform you that all comments have been adequately attended to as shown below. The authors wish to inform you that the corrections pointed out by the reviewers were effected and highlighted in yellow in the main manuscript and also in the responses below.

Thank you.

Eresanya E.O.

Corresponding author

Reviewers'	Comments	to	Author:
Reviewer:			1
Abstract	page.	Line	3
The author should correct the statement 'the land surface over Osogbo'. it is an incorrect statement.			
Response – This has been corrected.			

“The progressive nature of urbanization plays a prominent role in land-atmosphere processes which has corresponding impacts on the general environment. This research investigated the changing patterns of the **land use land cover over Osogbo** and its environs using remote sensing data obtained from Landsat TM, ETM+ and OLI/TIRS sensors. The changes in four land use classes were assessed **for 1984, 2000 and 2015**. The land surface temperature (LST) of the area was estimated from the satellite images covering the study periods and the Surface Urban Heat Island (SUHI) process were also investigated between the city of Osogbo and the surrounding towns. The results showed major urban expansion leading to urban sprawl within the vicinity. Urban area increased by 5106 hectares while vegetation decreased by 8653 hectares between 1984 and 2015 indicating major variations in the land surface features. This was revealed in the increase of the LST over the locations which ranged between 22.6 °C and 30 °C (mean 25.2 °C) in 1984 and between 29.3 °C and 36.7 °C (mean 31 °C) in 2015. The highest SUHI intensity was observed between the major urban area (Osogbo) and least developed towns. The continuous increase in surface temperature of the environment due to the continuous variations in the land surface properties implies increased risk of heat-related environmental issues such as deterioration of thermal comfort conditions.

line 5

the author should explain what he mean by this Thirty-one-year period because he only perform analysis on 1984, 2000 and 2015 year one after the other.

Response – The sentence has been modified to “The changes in four land use classes were assessed **for 1984, 2000 and 2015**”

The author is expected to use keywords not abbreviation in the keyword section.

Response – This has been corrected and the keywords included as “Urban area, Land Use Land Cover, Land Surface Temperature, Urban sprawl, Surface Urban Heat Island”

Page 5

I expected that the map projection in fig 1 should reveal climatological classification at local scale of the study area.

Response – the study area falls within just one climatological zone. This has been mentioned in the description of the area in section 2.1-**According to Köppen classification, it falls under the tropical wet and dry climate (Aw)**

“This research is carried over the region of Osogbo ($7^{\circ}48''N$ $4^{\circ}35''E$), Osun state, Nigeria (figure 1). Osogbo, the capital of Osun state located in the South-western region of Nigeria. Osogbo city has boundary with Ede, Olorunda, Obokun, Egbedore, Orolu, Ifelodun, Boripe, Atakumosa and Ilesa. **According to Köppen classification, it falls under the tropical wet and dry climate (Aw).** The climate is influenced by the Maritime Tropical (mT) Air mass, the Continental Topical (cT) Air mass and the Equatorial Easterlies. The average annual temperature is $26.1^{\circ}C$ and the annual precipitation is 1241mm, which falls mostly in summer. The driest and wettest month are January and September with 9mm and 202 mm values respectively while the warmest and coldest month are March and August with an average $28.3^{\circ}C$ and $23.7^{\circ}C$. Precipitation varies with 193cm between the driest and the wettest month. The variation in annual temperature is about $4.6^{\circ}C$. Osogbo is the Administrative and commercial center of Osun State. Osogbo has been witnessing tremendous increase in population since the creation of Osun state but this city has witnessed increase in road expansion and construction since the advent of the present government (2011 till date) which had led to increase urbanization and population inflow into Osogbo and its suburb”.

Page 8, 12 & 13

There are a lot of repetition here. the author should rewrite these pages to avoid repetition.

Response – The repetitions had removed and the sentences rearranged in a better format. **Kindly note that the corrections were highlighted with yellow colour as presented in section 3.1, 3.2 and 3.3.**

There is no acknowledgement refer to in this work.

Response – The acknowledgement section has been included immediate after the conclusion section.

Acknowledgements The authors will like to appreciate the United States Geological Survey (USGS) for the provision of the dataset used in this research work”

There are also citations that were not inline with the write up

Response – Those citation that were not inline with the write up had been removed.

These citations “Akinyoola, J.A., Eresanya, E.O., Orimoogunje, O.O.I. et al (2018): Monitoring the Spatio-Temporal Aerosol loading over Nigeria. Modeling Earth Systems and Environment. <https://doi.org/10.1007/s40808-018-0485-2>,

Osinowo, A.A., Balogun, I.A., Eresanya, E.O. (2018): Assessment of wave energy resource in the mid-Atlantic based on 37-year numerical hindcast data. Modeling Earth Systems and Environment. <https://doi.org/10.1007/s40808-018-0484-3> and

Osinowo A.A., Okogbue E.C., Eresanya E.O. et al. (2017): Evaluation of wind Potential and its trends in the Mid-Atlantic. Model. Earth Syst. Environ. 3:45. <https://doi.org/10.1007/s40808-017-0399-4>” were removed.

Reviewer: 2

Comments to the Author(s)

The study aim is to present the Research on “Investigation of the changing patterns of the land surface over Osogbo and its environs” The manuscript is presented clearly and nicely. However, to have scientific merit, the following comments should be addressed. I would like to suggest Major revision for this paper.

1. The introduction is too much. Better to reduce it.

Response – The introduction has been reduced and modified as can be seen in the introduction section on page 2.

2. Figure 1 = pls add a scale bar and African continent for Figure 1

Response – The scale bar has been added. This can be seen in the figure below:

Figure 1. Map of the study region showing the towns considered in the study

3. Pls add the time of the three images (Table 1)

Response – The time of the three images had been added as seen in **Table 1** highlighted in yellow

Table 1: Details of the satellite images used for the research

Satellite	Sensor	Path	Row	Source	Date acquired	Resolution (m)	Time (West African)
Landsat5	TM	190	55	USGS	11 th December, 1984	30 120 (30)	10:26
Landsat7	ETM+				15 th February, 2000	30 60 (30)	10:49
Landsat8	OLI/TI RS				17 th December, 2015	30 100 (30)	10:56

4. What are the points behind the selection of the satellite images (Table 1)

Response – The satellite images were selected in order to assess the changes in the land use land cover over Osogbo city from one interval. These periods revealed the changes in the land use land cover over the research area.

5. How did you measure the accuracy assessment of the LULC? Add details related to accuracy assessments.

Response – The accuracy assessment was done for the classified images. It was achieved using the kappa coefficient through the generation of 140 random points across the images. The points were then compared with referenced dataset from Osogbo.

6. Add some more related past research related to the UHI intensity in the African region. The calculation is not clear. There is two main methods used to classify the UHI intensity. Refer to the latest paper for your understanding. Read these papers for more knowledge.

- Spatial Analysis of Surface Urban Heat Islands in Four Rapidly Growing African Cities
- Impact of Landscape Structure on the Variation of Land Surface Temperature in Sub-Saharan Region: A Case Study of Addis Ababa using Landsat Data (1986–2016)
- Impact of Urban Surface Characteristics and Socio-Economic Variables on the Spatial Variation of Land Surface Temperature in Lagos City, Nigeria
- Quantifying Surface Urban Heat Island Formation in the World Heritage Tropical Mountain City of Sri Lanka

Response – More related researches as suggested by the reviewer had been added and referenced as stated below:

“ Babalola, OS. and Akinsanola, AA. (2016). Change Detection in Land Surface Temperature and Land Use Land Cover over Lagos Metropolis, Nigeria. *J Remote Sensing & GIS*, 5, 3. doi:10.4172/2469-4134.1000171

Balogun I.A, Daramola M.T. (2019a): The outdoor thermal comfort assessment of different urban configurations within Akure City, Nigeria. *Urban Climate* 29 (100489). <https://doi.org/10.1016/j.uclim.2019.100489>

Balogun I.A, Daramola M.T. (2019b): The impact of urban green areas on the surface thermal environment of a tropical city: a case study of Ibadan, Nigeria. *Spatial Information Research* 27(1) 23-36. <https://doi.org/10.1007/s41324-018-0219-6>

Barat A., Kumar S., Kumar P., Sarthi P.P. (2018): Characteristics of Surface Urban Heat Island (SUHI) over the Gangetic Plain of Bihar, India. *Asia-Pac. J. Atmos. Sci.*, 54(2), 205-214

Daramola, M.T., Balogun, I.A. (2019a). Analysis of the urban surface thermal condition based on sky-view factor and vegetation cover. *Remote Sensing Applications: Society and Environment* 15, 100253. <https://doi.org/10.1016/j.rsase.2019.100253>.

Dissanayake, D.M.S.L.B., Morimoto, T., Murayama, Y., Ranagalage, M. (2019). Impact of Landscape Structure on the Variation of Land Surface Temperature in Sub-Saharan Region: A Case Study of Addis Ababa using Landsat Data (1986–2016). *Sustainability* 11, 2257.

Dissanayake, D.M.S.L.B., Morimoto, T., Murayama, Y., Ranagalage, M., Handayani, H.H. (2018). Impact of Urban Surface Characteristics and Socio-Economic Variables on the Spatial Variation of Land Surface Temperature in Lagos City, Nigeria. *Sustainability* 11, 25.

Ranagalage, M., Dissanayake, D.M.S.L.B., Murayama, Y., Zhang, X., Estoque, R.C., Perera, E.N.C., Morimoto, T. (2018). Quantifying Surface Urban Heat Island Formation in the World Heritage Tropical Mountain City of Sri Lanka. *International Journal of Geo-Information* 7, 341.

Ranagalage, M., Dissanayake, D.M.S.L.B., Murayama, Y., Zhang, X., Estoque, R.C., Perera, E.N.C., Morimoto, T. (2018). Quantifying Surface Urban Heat Island Formation in the World Heritage Tropical Mountain City of Sri Lanka. *International Journal of Geo-Information* 7, 341.

Simwanda, M, Ranagalage, M, Estoque, R.C., Murayama, Y. (2019). Spatial Analysis of Surface Urban Heat Islands in Four Rapidly Growing African Cities. *Remote Sensing* 11, 1645 ”

7. Figure 2 and 6 – No scale bar or north arrow.

Response – Scale bar and the north arrow has been added as seen in the figures below (Figure 2 and 6)

Figure 2: Land use land cover map of the region of study for (a)1984 (b)2000 (c) 2015

Figure 6. Land surface temperature ($^{\circ}\text{C}$) for (a) 1984 (b) 2000 and (c) 2015

8. Figure 6 – use the same legend for three maps, and it will help to compare the changes

Response – separate color scales were used as a result of the large difference in the range for both periods. If identical color scales are used, the spatial visualization would be distorted. This is why the authors opted for the legends as it helps with visual representation of each year.

9. Figure 7 – witch locations.

Response – This has been corrected as shown in the Figure's title "Figure 7. Land Surface Temperature over Osogbo and the surrounding towns"

10. Better to divide result and discussion into Result and discussion. All result need to discuss in the result sections.

Response – The paper is now divided into the Results (3.1, 3.2 and 3.3) and Discussion (4.1 and 4.2).

3. Results and discussion

3.1. Land use land cover analysis

The land use maps of 1984, 2000 and 2015 are shown in Figure 2. As at 1984, the only noticeable urban areas were Osogbo, Ede and the Ilobu axis with Osogbo been the largest urban area. In 2000, these urban areas witnessed increase as the area expanded, however the most noticeable observation is the increase in urban areas in 2015. At this period, the areas experienced dramatic increase as the expansion occurred, and other locations previously rural became urbanized. The expansion of Osogbo and Ilobu axis was such that urban sprawl, which is the basically defined as the spreading of urban developments, occurred between 1984 and 2015.

The accuracy of the classified images was assessed and the result is presented in Table 3. The overall accuracy of the classification was 84.2% for 1984, 84.35% for 2000 and 84.14% for 2015. The kappa coefficient was 0.803, 0.812 and 0.806 for 1984, 2000 and 2015 respectively, the result reveals reasonable classification accuracy.

In order to explain and show the changes in the land use land cover over Osogbo and its environs, the area coverage (hectares) of the land cover classes were analyzed and plotted as shown in figure 3. In 1984, vegetation occupied majority of the land cover with an area coverage of about 43,000 hectares, barren land was about 32,500 hectares while water and urban area were about 340 and 3,200 hectares respectively. In 2000, vegetation reduced to about 39,000 hectares, while the barren land and urban area increased to about 35,300 and 4800 hectares respectively. In 2015, the area coverage of vegetation was about 34,300 hectares, barren land was about 36,200 hectares and urban area was about 8,300 hectares.

Figure 4 reveals the percentage cover of each land use class for each of the year studied. As stated earlier, the area coverage of vegetation decreased over time, from 54.4 % in 1984 to 49.1 % in 2000 and finally 43.3 % in 2015. Urban and barren land increased from 4.0 % and 41.1 % in 1984 to 6.0 % and 44.8 % in 2000, and then to 10.5 % and 45.9 % in 2015 respectively. Water body was 0.4 %, 0.1 % and 0.2 % in 1984, 2000 and 2015 respectively. It is evident that

vegetation cover occupied majority of the land surface area within the region, however, the percentage decreased towards 2015, with barren land and urban increasing. Figure 3 also showed that, the expansion of the urban area cover which was accompanied by increase in barren land brought about conversion of the vegetated portion of the land use.

Figure 5 showed the change in area cover for each land use between the different intervals. Between 1984 and 2000, vegetation lost about 4,200 hectares to urban and barren land while urban and barren land gained 1,558 and 2,861 hectares respectively. From 2000 to 2015, vegetation lost another 4,456 hectares to other land use classes, while both urban and barren land gained 3,547 and 879 hectares respectively. The total loss of vegetation cover area over the region between 1984 and 2016 was 8,653 hectares while the total increase in urban and barren land were 5,105 and 3,739 hectares respectively.

3.2 Land Surface Temperature Analysis (LST)

This section presents the result of estimated Land Surface Temperature (LST) from satellite images over the region. The land surface temperature maps are presented in figure 6. The land surface temperature distribution clearly shows the highest LST intensities over the urbanized areas of the region such as Osogbo, Ede and Ilobu axis. The lowest LST intensities are observed over the peripheries of the region designated as vegetation.

Figure 7 shows the land surface temperature over each of the locations considered in this study. Evident increase in the land surface temperature over the thirty-one-year period is presented in the chart. The LST intensity over all the locations increased significantly between 1984 and 2015 with a mean value of 5.8 °C. The highest increase within the period was over Osogbo with an increase of 6.8 °C. The mean LST increased from 26.2 °C in 1984 to 30.5 °C in 2015 (Table 4), indicating a progressive warming of the region.

3.3 Surface Urban Heat Island (SUHI)

Land Surface Temperature (LST) has been recognized as a pointer to the existence of surface urban heat island (SUHI). The surface urban heat island estimated from the LST over the region is presented figure 8. SUHI was noted across the towns for each of the years studied. The mean SUHI intensity showed that Dagbolu, Okinni and Aba Adi had the highest intensities while Ilobu had the lowest intensity. The towns with the highest SUHI intensities were the rural areas of the region with lower intensity of urbanization and associated human activities which bring about increased thermal intensity. Alternatively, Ilobu with the lowest intensity of SUHI, is a sub-urban area.

4. Discussion

4.1. Land Use Land Cover Change

The result from this study reveals remarkable changes in the land use land cover of Osogbo and its environs over the years studied. Agricultural activities are known to be the major activities of the resident of Osogbo and its environs in the past, about 68% of the land were used for agricultural related practices; but since it became the state capital in 1991, the area had experienced drastic change in land use. Most agricultural relevant lands had been altered to built-

up areas. Aside from the state capital institution of Osogbo in 1991, another noticeable factor responsible for urban spatial expansion was the establishment of Osogbo machine tools and steel rolling mills companies which played a significant and pivotal role in the economic activities of the capital city and caused development along the northern axis of the city. The study region was largely axial in its spatial expression, growing along major arteries of transportation. These axes of growth were separated by undisturbed forest, secondary vegetation and wetlands. This became most noticeable in 2000 which represents the beginning of noticeable development in the region, with rise in the built-up regions and reduction in vegetative coverage which is due to the massive influx of people from the suburb to the Capital. In 2000, the city of Osogbo experienced major developments such as, a new government secretariat built by the state government and also, other parastatals began to spring up such as House of Assembly Complex and Central Bank Branch along Gbongan-Ibadan axial road. Private residential and commercial developments were also springing up in response to the burgeoning population of mostly government workers and their dependents, resulting in low-density development towards the periphery.

This study corroborates with the findings of Samuel and Atobatele (2019) which established the fact that transition to civil rule yielded economic dividends which further intensified urban expansion, with concomitant accelerated loss of both vegetable cover and water bodies in 2000. The period equally marked the implementation of a wage increase of 15% in the country which afforded workers the opportunity to build their own houses. The economic fortunes of the people have significant effect on the trajectory of urban development coupled with the inflow of migrants from suburbs and the pastoral areas in search of government occupations and contracts in the newly created State. The sustenance of urban land use growth in this period is partly due to economic boom that led to the over saturation of the surrounding lower-order settlements such as Dagbolu, Kelebe, Ofatedo, Okinni, Abere, Ido Osun, Ilobu, Ifon and Iragberi.

Since then, the vacant land and wetlands within the urbanized areas were being developed, in response to increased demand from the government workers (both Federal and State) who had to move to the State capital as result of the creation of the State. Expectedly, the appreciable gain in urban expansion caused a concomitant reduction in both vegetation cover and water. As the residents in the city increase across the years, this is followed by a corresponding increase in the urbanized area with consequential reduction in vegetative area. Land Use Land Cover analysis of Osogbo in 2015 revealed that there is increase in development with the further expansion in built-up Facross the area, which is expected to continue due to the continuous rise in the number of its residents. The inflow of people from outside the urban areas to the urban centers in search of job opportunities and better standard of living leading to increased urban expansion has also been noticed in some of the capital cities in Nigeria such as Lagos (Babalola and Akinsanola, 2016) and Ibadan (Daramola et al., 2018), as well as other countries (Mohammady and Delavar 2016; MacLachlan et al., 2017; Son and Thanh, 2017).

By 2015, the area coverage of urban increased while vegetation reduced. During this period, the city appeared to have expanded to its traditional and administrative limits, as the development of hitherto undeveloped plots within the built-up areas, which had started earlier, intensified as identified by Aguda and Adegboyega (2013). It was at this period that the conversion of

vegetation reached its peak, with the consequent deleterious effect on the ecosystems services. The continued increase in the expansion of Osogbo city caused a spill over into other smaller sub-urban centres that were not considered as part of the city. However, adjoining towns such as Owode Ede, Ilobu, Erin Osun, Ifon Osun, Oba-Ile, Oba-Oke, Dagbolu, Ido Osun and Ede continued to receive the spill over population and development from Osogbo.

4.2. Effect of Land Use Land Cover Change on Land Surface Temperature

The changing land use land cover of Osogbo and the surrounding towns have effects on the surface thermal condition of the area. This was observed in the land surface temperature distribution and intensity for the years studied. The varying intensity of land surface temperature with land use land cover has been reported in studies across the globe (James and Charles, 2014; Pal and Ziaul, 2017; Estoque et al., 2017; Dissanayake et al. 2018; 2019). The intensity of LST decreased outwards from the urbanized areas to the areas with less urban surfaces as the highest intensities are within the urban core. The dispersion of LST intensity from the urban core to the peripheries of the region clearly shows the temperature differential between the urban area and other land use land covers such as vegetation. It can also be seen that, the intensity of land surface temperature increased over the area with time. Initially in 1984, the high LST intensity was concentrated over Osogbo, however with the expansion of urban developments outwards from the town center, the intensity increased over the altered land use areas. As stated earlier, Osogbo is the major urban area within the region and with the degree of urbanization, population and intensity of human induced activities, over the past couple of years, the surface temperature of the city increased significantly. This indicates the effect of urbanization on the thermal condition of the environment. According to Barat et al. (2018), rapid development of metropolitan areas accounts for intensification of the temperature of the land surface.

Urbanization and its associated activities, has brought about changes in the thermal situation of an area (Ogunjobi et al. 2018b, Eresanya et al., 2017, Daramola and Balogun 2019a; 2019b), greatly affecting the intensity of surface urban heat island, as factors such as impervious surfaces, intense human activities, reduced vegetation covers all bring about increase in the urban temperature (Dewan et al. 2012; Song et al. 2014; Meng et al. 2018). This consequently affects the thermal comfort condition of the region (Njoku and Daramola 2019; Balogun and Daramola 2019a). The subject of increased intensity of urban surface temperature has been reported in cities over the globe (Dewan and Corner 2014; Peng et al., 2016; Chen et al., 2017; Yu et al., 2018). Several mitigation strategies for the increased urban temperature have been considered, of which increased greenery of the cities has been proposed by several studies (Santamouris et al. 2011; Dou 2014; Doick et al., 2014). Vegetation within the urban environment such as forest reserves, urban parks, affects the surface temperature due to the reduction in surface heating by the incoming solar radiation (Armson et al., 2013; Yang et al., 2017; Xu et al., 2017). Vegetation cover such as trees, shield the land surface from direct heating and therefore brings about reduced temperature of the surface compared to that of impervious surfaces which characterizes the urban area.

Appendix B

Dear Editor,

Thank you for accepting our manuscript for publication in Journal of Royal Society Open Science. The authors wish to inform you that the comments had been adequately attended to and can be found in the responses below and also in the main documents in red color.

Thank you.

Eresanya E.O.

Corresponding author

- The authors information are as supply below:

Eresanya Emmanuel Olaoluwa^{1, 2}, Daramola Mojolaoluwa Toluwalase³, Durowoju Olufemi Sunday^{3,4}, and Awoleye Peace³

¹Department of Marine Science and Technology, Federal University of Technology, P.M.B 704, Akure, Nigeria

²South China Sea Institute of Oceanology, Chinese Academy of Chinese, 164 West Xingang Road, Guangzhou, 510301, P. R. China

³Department of Meteorology and Climate Science, Federal University of Technology, P.M.B 704, Akure, Nigeria

⁴Department of Geography & Resource Management, Osun State University, Okukum, Osun State, Nigeria

- **Ethics statement:** This research made use of Cloud free satellite data from the Landsat sensors TM, ETM+ and OLI/TIRS. The images were obtained from the United States Geological Survey (USGS).
- **Data:** Cloud free satellite data from the Landsat sensors TM, ETM+ and OLI/TIRS were used in this study. The images were obtained from the United States Geological Survey (USGS) for 1984, 2000 and 2016 covering a period of 32 years. Data can be accessed through <https://earthexplorer.usgs.gov/>.
- **Competing interests:** The authors declare that there is no competing interest.
- **Authors' contributions:** Eresanya conceived the study, designed the study, coordinated the analysis, Daramola and Awoleye participated in the preliminary data collection and analysis, Eresanya and Daramola carried out sequence alignments with further analysis and drafted the manuscript; Durowoju carried out the first draft of Literature review; Eresanya proofread the complete manuscript before submission. All authors gave final approval for publication.

- **Acknowledgment:** The authors will like to appreciate the United States Geological Survey (USGS) for the provision of the dataset used in this research work. We will also want to thank the anonymous reviewers who provide comments that substantially improved the manuscript.
- **Funding statement:** The authors wish to declare that there is no funding